# The Catalytic-Dependent and -Independent Roles of Lsd1 and Lsd2 Lysine Demethylases in Heterochromatin Formation in *Schizosaccharomyces pombe*

**DOI:** 10.3390/cells9040955

**Published:** 2020-04-13

**Authors:** Bahjat F. Marayati, James F. Tucker, David A. De La Cerda, Tien-Chi Hou, Rong Chen, Tomoyasu Sugiyama, James B. Pease, Ke Zhang

**Affiliations:** 1Department of Biology, Center for Molecular Signaling, Wake Forest University, Winston-Salem, NC 27109, USA; marab15@wfu.edu (B.F.M.); iamjamestucker@gmail.com (J.F.T.); ddelacer@wakehealth.edu (D.A.D.L.C.); hout217@wfu.edu (T.-C.H.); peasejb@wfu.edu (J.B.P.); 2Physiology and pharmacology, Wake Forest School of Medicine, Winston-Salem, NC 27101, USA; rchen@wakehealth.edu; 3School of Life Science and Technology, ShanghaiTech University, Shanghai 201210, China; tsugiyama@shanghaitech.edu.cn

**Keywords:** Lsd1, Lsd2, heterochromatin, lysine demethylase, gene silencing, H3K9me2

## Abstract

In eukaryotes, heterochromatin plays a critical role in organismal development and cell fate acquisition, through regulating gene expression. The evolutionarily conserved lysine-specific demethylases, Lsd1 and Lsd2, remove mono- and dimethylation on histone H3, serving complex roles in gene expression. In the fission yeast *Schizosaccharomyces pombe*, null mutations of Lsd1 and Lsd2 result in either severe growth defects or inviability, while catalytic inactivation causes minimal defects, indicating that Lsd1 and Lsd2 have essential functions beyond their known demethylase activity. Here, we show that catalytic mutants of Lsd1 or Lsd2 partially assemble functional heterochromatin at centromeres in RNAi-deficient cells, while the C-terminal truncated alleles of Lsd1 or Lsd2 exacerbate heterochromatin formation at all major heterochromatic regions, suggesting that Lsd1 and Lsd2 repress heterochromatic transcripts through mechanisms both dependent on and independent of their catalytic activities. Lsd1 and Lsd2 are also involved in the establishment and maintenance of heterochromatin. At constitutive heterochromatic regions, Lsd1 and Lsd2 regulate one another and cooperate with other histone modifiers, including the class II HDAC Clr3 and the Sirtuin family protein Sir2 for gene silencing, but not with the class I HDAC Clr6. Our findings explore the roles of lysine-specific demethylases in epigenetic gene silencing at heterochromatic regions.

## 1. Introduction

In eukaryotes, DNA is packaged with its associated proteins, forming chromatin. Nucleosomes refer to the basic units of chromatin; 147 base pairs of DNA wound around a histone core, made up of two copies each of histones H2A, H2B, H3, and H4 [1]. Histones play important roles in many cellular processes, including transcription, repair, recombination, and higher order chromatin organization [2,3]. The N-terminal tails of these histones are unstructured, radiate out from the nucleosome core, and are subject to covalent modifications, such as methylation, phosphorylation, and acetylation by chromatin-modifying enzymes [4]. These modifications adjust the density of chromatin, the dynamic associations between adjacent nucleosomes, and the recruitment of various factors, all of which control the accessibility of transcriptional machinery and gene expression [5,6,7,8]. Differences in histone modifications give rise to two distinct phases of chromatin: euchromatin and heterochromatin [9]. Euchromatin describes the less condensed regions of chromatin, associated with active transcription and recombination [10]. Heterochromatin, on the other hand, describes regions of highly condensed chromatin with low levels of transcriptional activity and is thus considered to be a repressive, or silenced, chromatin state [8].

In the fission yeast *Schizosaccharomyces pombe* (*S. pombe*), heterochromatin is constitutively enriched in centromeres, telomeres, and the mating-type locus; each of which contains repetitive elements serving as nucleation points for heterochromatin formation [11]. Nucleation of heterochromatin at the centromere is dependent on the transcription of *dg/dh* repeat regions by RNA polymerase II (RNAPII) and the subsequent processing of these transcripts into siRNAs by RNA interference (RNAi) machinery [11,12,13]. Briefly, RNAPII transcripts from these repeats are converted to double-stranded RNAs by the RNA-dependent RNA polymerase complex (RDRC) [14], processed by dicer into siRNAs [15,16], and then loaded onto the argonaute-containing RITS complex [14]. The RITS complex is targeted to these repeat regions through the homology of the siRNA sequence [17], which associates with chromatin by direct interaction with H3K9me [18,19,20], then recruits the Clr4 complex (ClrC) to initiate chromatin remodeling [19,21]. Once associated with chromatin, the RITS complex reinforces silencing through the direct recruitment of the RDRC, initiating the synthesis of new siRNAs for positive feedback [22,23]. This mechanism is well conserved in plants and worms, as the assembly of large heterochromatic domains requires dicer-dependent argonaute localization [24,25,26].

The nucleation of heterochromatin at the mating-type locus of *S. pombe* also occurs in an RNAi-dependent manner through the processing of transcripts produced from the *cenH* region, which is homologous to the *dg/dh* repeats of the centromere [27]. Unlike peri-centromeric regions, this process is only partially dependent on RNAi, as low efficiency heterochromatin assembly at the mating-type locus still occurs in the absence of RNAi or *cenH* and has been attributed to a parallel pathway involving transcription factors Atf1 and Pcr1 [27]. In addition to RNAi, the RNA degradation exosome complex has been implicated in a parallel, RNAi-independent mechanism at the centromere. In the absence of RNAi machinery, ClrC can still be recruited to nucleate heterochromatin. This process may rely on the accumulation of non-coding RNAs produced from centromeric repeat regions [28], and the degradation by 5′-3′ exoribonuclease Dhp1/Xrn2 [29,30].

H3K9 methylation by Clr4, a critical early step in the formation of heterochromatin, is a progressive reaction that competes with H3K9 acetylation. Therefore, the removal of acetyl groups from histone tail lysines by histone deacetylases (HDACs), such as Clr3 and Sir2, provides the substrate to Clr4, which is necessary for heterochromatin assembly [31]. The deacetylation of histone tails facilitates the condensation of chromatin by directly affecting the interactions between nucleosomes [5]. As such, HDACs have received increasing attention in recent years as critical mediators of the nucleation, spreading, and maintenance of heterochromatin. Clr3 and Sir2 appear to have overlapping but distinct roles in the establishment and spreading of heterochromatin at the centromere [31,32]. Additionally, the RNAi-independent maintenance of heterochromatin also appears to be dependent on Clr3 and Sir2 [32]. A third HDAC in *S. pombe*, Clr6, has been shown to be primarily involved in the repression of euchromatic loci via deacetylation of promoter regions [33,34], although it also controls the transcription of repetitive regions including *dg/dh* repeats and retrotransposons [35,36].

The removal of methyl groups from histone tails is mediated by conserved amine oxidase- and Jumonji C (JmjC) domain-containing enzymes known as histone demethylases, although less is known about their roles in heterochromatic silencing [37,38,39]. Epe1 is a JmjC domain protein and putative histone demethylase that has been shown to act as an anti-silencing factor, limiting the spreading of heterochromatin to appropriate functional boundaries and countering the propagation of heterochromatin over multiple rounds of cell division [37,40,41,42,43,44]. However, another JmjC domain protein, Lid2, interacts with H3K4 methyltransferase Set1 and H3K9 methyltransferase Clr4 to coordinate H3K4 and H3K9 methylation and functions as a pro-silencing factor [45]. While it has been widely accepted that HDACs are important for heterochromatin formation [31,46,47], and more attention has been drawn towards the roles of histone demethylases in recent years, the exact mechanisms by which these enzymes contribute to the different stages of heterochromatin assembly are still under investigation.

LSD1/KDM1a is a highly conserved lysine-specific demethylase that controls the expression of numerous loci, by targeting the demethylation of mono- and dimethylated histone H3 (K4 or K9) [39,48,49]. Lsd1 can act as a transcriptional repressor or activator, depending on the specificity and dynamics of its associating proteins. For instance, when mammalian LSD1 is associated with androgen receptor (AR), it specifically targets H3K9 for demethylation, leading to the de-repression of AR target genes [48]. In contrast, when recruited by a SANT domain-containing co-repressor CoREST, LSD1 demethylates H3K4 on nucleosome substrates, negatively regulating transcription [50,51,52]. Additional transcriptional repression by LSD1 is mediated through its interaction with other repressive complexes including NRD (nucleosome remodeling and deacetylating complex), CtBP and HDAC complexes [53,54,55,56].

LSD2/KDM1b, the mammalian paralog of LSD1, also shows dual specificity for H3K9 and H3K4 demethylation [57,58,59], however LSD2 appears to perform distinct functions from LSD1. For example, unlike LSD1, LSD2 does not form stable associations with CoREST [60]. In addition, LSD1 primarily localizes to promoter regions, while LSD2 binds to gene bodies [59]. While the biological roles of LSD2 are beginning to be appreciated, much less is currently known about the function of LSD2 than LSD1 [56]. The multifaceted functions of both LSD1 and LSD2 highlight the complexity of understanding their roles in chromatin regulation and in their coordination with other chromatin modifiers.

*S. pombe* contains orthologs of both LSD1 and LSD2, which are missing in budding yeast [61]. Lsd1 copurifies with Lsd2 and plant homeo-domain finger proteins Phf1 and Phf2, forming the Lsd1/2 complex, but does not appear to form stable associations with HDACs, unlike human LSD1 [34,49,61,62]. Lsd1 is required for efficient growth in *S. pombe* and plays roles in the suppression of antisense transcripts and boundary regulation [34,49,61]. Although *S. pombe* Lsd1 and Lsd2 localize to the heterochromatic regions, they do not appear to target H3K4 for demethylation at these regions, and only specifically demethylate H3K4 at euchromatic promoter regions [49]. Since *lsd2^+^* is essential and is not amenable to functional studies by complete deletion, the function of Lsd2 has been largely unexplored in fission yeast. Interestingly, catalytically inactive mutant strains of *lsd1^+^* and *lsd2^+^* (*lsd1-ao* or *lsd1-KK603-604AA*; *lsd2-ao* or *lsd2-KK861-862AA*) are both viable and do not show significant growth defects [39,49,63]. These findings indicate that Lsd1 and Lsd2 perform functions essential for viability beyond their known histone demethylation activities. So far, their histone demethylation-independent roles have not been explored in *S. pombe*. Furthermore, the reported viability of *lsd1Δ*, but not *lsd2Δ*, suggests that Lsd2 performs additional, essential functions. To what degree Lsd1 and Lsd2 act independently is not yet understood.

In this study, we explore the functions of Lsd1 and Lsd2 in epigenetic silencing of constitutive heterochromatic regions in *S. pombe*, through the generation of novel Lsd mutants. We identify an N-terminal peptide that is essential for the nuclear localization of Lsd2. Although the catalytic mutants of either *lsd1* or *lsd2* alleviate silencing defects caused by the loss of *ago1^+^*, C-terminal truncated mutants of *lsd1* or *lsd2* show cumulative defects with *ago1Δ* in gene silencing at all main heterochromatic regions. These findings suggest that Lsd1 and Lsd2 function to repress heterochromatic transcripts at heterochromatic regions, partially through a mechanism independent of amine oxidase-related demethylation activities. In addition, without the amine-oxidase activities of Lsd proteins, cells are defective in the RNAi-independent maintenance and re-establishment of heterochromatin at the mating-type locus. We also provide evidence supporting a model, in which Lsd1 and Lsd2 regulate one another and serve additional, cooperative roles with the other heterochromatic factors, including histone deacetylases Clr3 and Sir2.

## 2. Materials and Methods

### 2.1. Yeast Strains and Cell Culture

The *S. pombe* strains used in this study are listed in Appendix A and genotypes were confirmed using primers listed in Appendix A. Cells were cultured using standard laboratory procedures for growth and manipulation [64]. Epitope-tagged and deletion mutant strains were generated using the standard PCR methods described previously [65]. Genetic crossing followed by tetrad dissection were used to generate double and triple mutant strains.

### 2.2. Dilution Assay

Liquid cultures were diluted in series (1:10) and plated using a pin transfer tool on YEA media (rich). All cultures were grown at 30 °C (or at 37 °C where indicated) for 2–3 days until single colonies appeared in the most diluted spot.

### 2.3. RNA-Seq and ChIP-seq

RNA was isolated from *S. pombe* strains (each genotype represented by 3 independently derived biological replicates), using the PureLink RNA Mini Kit (Life Technologies, catalog#: 12183025, California, USA) and TRIzol Reagent (Invitrogen Corp, Catalog#: 15596026, Massachusetts, USA), following manufacturer’s protocols. RNA was recovered in RNase-free water, frozen, and shipped to BGI Americas (Cambridge, Massachusetts, USA), where the RNA-seq quantification (BGISEQ-500 platform; 50 bp single-end; 20M clean reads) with standard bioinformatics service was performed. Each of the 50 bp single-end RNA-Seq reads for each sample (wild-type, *lsd1-∆HMG*, *lsd2-∆C*) were aligned to the *S. pombe* reference genome ASM294v2 using the STAR 2-pass method [66,67], with an average of 2.5 × 10^7^ uniquely mapped reads. Aligned reads were quantified using FeatureCounts [68] with options -*t* gene -*o* fraction, which allowed for the quantification of an average of 2.5 × 10^7^ reads per sample to be assigned to the ASM294v2 genomic feature file. A differential gene expression (DGE) analysis was performed using limma with the -voom extension with Empirical Bayes smoothing of gene-wise standard deviations [69]. The top DGE genes were considered to be those where *p* ≤ 0.001 and comparisons were made between wild-type and *lsd1-∆HMG* as well as wild-type and *lsd2-∆C*. Further assessment was done by filtering log_2_ fold-change differences either ≥2 or≤ −2, in order to validate current findings to those of previous reported [39]. RNA-seq data in this study were deposited in the gene expression omnibus with accession number GSE148191. The wild-type H3K9me2 ChIP-Seq data were published previously and can be accessed publicly in the gene expression omnibus, with accession number GSE119604 [70].

### 2.4. qRT-PCR

Total RNA was prepared using the MasterPure Yeast RNA Purification Kit (Epicentre by Lucigen Corp, catalog#: MPY03100, Wisconsin, USA). First-strand cDNA synthesis was performed with M-MLV Reverse Transcriptase (Promega Corp, catalog#: M1701, Wisconsin, USA) following manufacturer protocols. Quantitative PCR was performed on a QuantStudio 3 Real-Time PCR system (Applied Biosystems, catalog# A28567, California, USA) with SYBR Select Master Mix (Applied Biosystems, catalog#: 4472920, California, USA). First-strand cDNA synthesis without reverse transcriptase served as negative controls. At least two biological replicates were performed for all experiments. Samples were triplicated for qPCR. Statistical analysis was performed using a student’s *t* test (two-tailed distribution). Error bars represent standard error of the mean (*s.e.m.*).

### 2.5. Chromatin Immunoprecipitation (ChIP)

ChIP experiments were performed as described previously, using an antibody against Histone H3 (di-methyl K9) (Abcam, catalog#: ab1220, Cambridge, UK) [11,29]. Histone H3 (di-methyl K9) ChIPs that were performed in FTP (Protein A-, TEV cleavage site, and Flag- double epitope-tag) backgrounds were modified. After sonication and fragmentation of chromatin, whole cell extracts were incubated with 50 units of AcTEV protease (Invitrogen Corp. Catalog#: 12575015, Massachusetts, USA) and 50µL IgG sepharose (GE Healthcare, Catalog#: 17096901, Illinois, USA) at 4 °C overnight. The next morning, pre-cleared whole cell extracts were collected after centrifugation at 17,530× *g* for 10 min, followed by adding the antibody (H3K9me2, Abcam ab1220). Quantitative PCR was performed on a QuantStudio 3 Real-Time PCR system (Applied Biosystems) with SYBR Select Master Mix (Applied Biosystems). At least two biological replicates were performed for all experiments with triplicate technical replicates for qPCR analysis. Statistical analysis was performed using a student’s *t* test (two-tailed distribution). Error bars represent standard error of the mean (*s.e.m.*).

### 2.6. Fluorescence Microscopy

Calcofluor stain was performed as previously established [71]. Hoechst stain for GFP live-imaging was followed as shown formerly [72]. Cells were imaged on a Zeiss 880 LSCM (Figure 1C and Appendix A). Results from biological duplicates were combined for final results.

### 2.7. Heterochromatin Formation and Maintenance Assay Using an ade6^+^ Reporter

The original *KΔ::ade6^+^* strain was generated previously [29]. Cells that carry the *KΔ::ade6^+^* off epiallele and formed red colonies on low adenine medium (YE) were selected for genetic crosses. All strains carrying *KΔ::ade6^+^* off in different genetic backgrounds were made by genetic crossing. After genotyping, colonies were amplified on rich medium and then stored in −80 °C. For heterochromatin formation and maintenance assays, stored cells were recovered from −80 °C, grown at 30 °C for 2 days, then were used for RNA extraction and RT-PCR (Appendix A). Individual cells were plated on YEA plates using a dissection microscope and were grown at 30 °C until small colonies appeared (4 days for *lsd1/2* C-terminal mutants or 2 days for other strains). The YEA plates were then replica-plated on YE plates and grown for an extra 2–4 days until colony color could be observed (Figure 5B). White (W) or red (R) colonies from YE plates were diluted in the water, and individual cells were dissected on YEA plates and grown for 2–4 days until small colonies appeared, then were plate-replicated on YE plates and grown for extra 2–4 days until the pigments of colonies could be assessed. Around 60 colonies for each genetic background were documented, and the percentage of colored and white colonies were calculated (Figure 5C). Individual red, white, and color-mixed colonies were cultured to exponential phase growth in liquid YEA, before being collected, either for RNA extraction (Figure 5D) or ChIP (Figure 5E).

### 2.8. Growth Curve

Growth curves were generated using a plate reader, as described previously [73]. Strains with indicated genotypes (Figure 2B and Figure 4A,B) were recovered from −80 °C, and grown on YEA plates at 30 °C for 2 days. Three replicates of each strain were inoculated in 1mL of liquid YEA, and the optical density (OD, 600nm) was measured. All strains were then diluted in liquid YEA to 0.1 OD and resuspended in wells to a total of 200 µL in an optical flat-bottom 96-well plate (Falcon). Cell growth was monitored using a microplate reader (Synergy H1 Hybrid Multi-Mode Reader, Biotek Instruments, Vermont, USA) with an incubation temperature of 30 °C, continuous fast orbital shaking, and OD (600 nm) readings every 2 min for 24 h. Growth curves for individual colonies were generated and analyzed by Gen5 microplate reader software (Biotek).

## 3. Results

### 3.1. A N-Terminal Region of Lsd2 Is Required for Nuclear Localization and Viability

Since the essential *lsd2^+^* cannot be completely deleted, we generated viable, partially functional mutant alleles for mechanistic studies. While Lsd1 and Lsd2 are structurally similar, Lsd2 has a unique N-terminal section (M1-P376) that contains an intron and is absent in Lsd1 (Figure 1A). We generated multiple N-terminal truncated mutants of Lsd2 and assessed their relative viability and growth defects (Figure 1B and Appendix A). The removal of large sections of the N-terminus resulted in either no or very minor growth defects, as seen in *lsd2-N1* (*lsd2*ΔQ157-P376) and *lsd2-N2* (*lsd2*ΔE57-P376) (Appendix A). The complete removal of the N-terminus resulted in lethality, as seen in *lsd2-N3* (*lsd2Δ*M1-P376) (Figure 1B). The lethality of this deletion indicates an essential function of this N-terminal region.

Based on the vastly different growth phenotypes observed, we generated additional mutants by removing more amino acids from the N-terminus. *lsd2-N4* (*lsd2*ΔG48-P376) was viable and showed only mild defects in growth (Figure 1A and Appendix A). Additional removal of the following 13 amino acids immediately upstream (_35_SFYSMNTSENDPD_47_) was catastrophic, as *lsd2-N5* (*lsd2*ΔS35-P376) was confirmed inviable (Figure 1B). We were unable to generate strains in which small regions proximal to the intron near the start codon (the exon are spliced together to form the W8 codon of Lsd2) were removed, suggesting that these truncations may interrupt the splicing of *lsd2^+^* or affect its nuclear localization. The confocal imaging of heterozygous diploid strains with *GFP*-tagged *lsd2-N3*, *lsd2-N4*, and *lsd2-N5* reveals that only the *lsd2-N4-GFP* localizes to the nucleus, while *lsd2-N3-GFP* and *lsd2-N5-GFP* do not. Therefore, the 13 amino acid sequence (_35_SFYSMNTSENDPD_47_) near the N-terminal region contains essential residues that govern the nuclear localization of Lsd2 and are required for viability (Figure 1C).

### 3.2. The HMG-Box Domains Are Essential for Lsd1 and Lsd2 Functions

The featured domain organization of Lsd1 and Lsd2 in *S. pombe* includes a SWIRM domain, an amine oxidase (AO) domain, and an HMG-box (Figure 1A and Figure 2A). The high-mobility group (HMG)-box domain proteins play essential roles in recognizing DNA or nucleosomes during various DNA-dependent processes including transcription, replication, and repair [74]. Both Lsd1 and Lsd2 contain C-terminal-HMG-box domains, but their functions in *S. pombe* have not been studied sufficiently. The N-terminal mutant strains of *lsd2^+^* were either inviable or showed minimal growth defects (Figure 1 and Appendix A), and *lsd1Δ* had severe growth defects. Therefore, we endeavored to generate additional mutants that are more suitable for functional studies. Lsd1 and Lsd2 mutants with intermediate growth defects indicative of functional abnormalities were produced by deleting portions of the C-terminus. We separately truncated the C-terminal HMG-box domain (*ΔHMG*) and just the C-terminal region immediately downstream of the HMG-box domain (*ΔC*; retaining the HMG domain) (Figure 2A). Haploid *lsd1-ΔHMG* (*lsd1Δ*841-1000) and both *lsd1-ΔC* (*lsd1Δ*934-1000) and *lsd2-ΔC* (*lsd2Δ*1203-1273) alleles are viable and amenable to laboratory manipulation (Figure 2B and Appendix A). However, deletion of the HMG domain was catastrophic for Lsd2, as *lsd2-ΔHMG* (*lsd2Δ*1113-1273) is inviable (Appendix A). Additionally, *lsd1-ΔHMG* and *lsd2-ΔC* show defects at the mating-type locus (Appendix A), consistent with the previously reported observation that the HMG domain of Lsd1 is required for effective imprinting at the mating type locus [63].

While *lsd1-ΔHMG*, *lsd1-ΔC*, and *lsd2-ΔC* were viable as haploid cells, they did form smaller colonies than wild-type cells under the same growth conditions, suggesting growth defects (Appendix A). Since the catalytic mutants of Lsd1 and Lsd2 (*lsd1-ao* and *lsd2-ao*) barely show growth defects [39,49,63], we investigated whether processes other than lysine demethylation were affected in these C-terminal mutants. We compared the doubling time of wild-type, catalytic, and C-terminal truncated mutants of *lsd1* and *lsd2* in liquid media (Figure 2B). We found that *lsd1-ΔHMG* and *lsd2-ΔC* have intermediate growth defects, and are therefore more applicable for laboratory manipulation. Further dilution assays demonstrated that the catalytic mutants of Lsd1/2 show wild-type-like growth and resistance to heat stress, but C-terminal mutants have varying degrees of growth defects and heat sensitivities on solid rich media (Appendix A). The analysis of homozygous and heterozygous diploids by the dilution assay revealed that the growth defects and heat sensitivities of these mutants are not dominant negative effects (Appendix A).

To examine cell morphology, we next observed these mutant strains via confocal fluorescence microscopy with calcofluor white staining (Appendix A). Consistent with dilution assay results, catalytic *lsd1-ao* and *lsd2-ao* mutants appeared much like wild-type, with no obvious defects in morphology. *lsd1-ΔHMG* cells, however, displayed prominent defects such as elongated cells, branching cells, and cells with multiple septa, indicating defective cell division. While *lsd1-ΔC* appeared like wild-type in morphology, the *lsd2-ΔC* mutant showed similar defects as *lsd1-ΔHMG*, although not quite to the same degree, with long or branched cells commonly observed. Consistent with our mutant phenotypes, *lsd1Δ* mutants were reported to display flocculation, morphological irregularities, and multi-nucleate phenotypes [34,39]. The results indicate that the HMG-box domain of Lsd1 is important for proper cell growth and morphology.

### 3.3. Lsd1 and Lsd2 C-Terminal Mutants Affect Genome-Wide Gene Expression and Display Defective Silencing at Constitutive Heterochromatic Regions

As Lsd2 is essential, transcriptome analysis in *S. pombe* has been limited to the loss of Lsd1 function. To explore the function of Lsd2 in differential gene expression (DGE), we analyzed the transcriptomes of *lsd2*-*ΔC* mutants using RNA-sequencing (RNA-seq). We also performed RNA-seq in *lsd1-ΔHMG* to compare with *lsd2*-*ΔC*, as both are novel Lsd mutants. When compared with previously published expression data in *lsd1Δ* [39], our data agree with the expression profiles for the 68 highest-confidence target genes of Lsd1 (Appendix A and Appendix A). As expected, DEGs (both up- and down-regulated) in *lsd2*-*ΔC* mutant are highly similar to those of *lsd1-ΔHMG* mutants (*r*^2^ =0.904) (Figure 2C). This result indicates widely overlapping functions between the two proteins. We observed more up-regulated than down-regulated genes (using a 2- fold cut-off) in both *lsd1-ΔHMG* and *lsd2*-*ΔC* mutants, suggesting that the dominant roles of Lsd1 and Lsd2 involve the repression of gene expression (Appendix A). Previous studies have identified the loss of heterochromatic silencing and a moderate drop in heterochromatin-associated H3K9me2 levels in *lsd1Δ* [39,49], We also observed a loss of silencing at the peri-centromeric, mating-type, and sub-telomeric regions in *lsd1-ΔHMG* and *lsd2*-*ΔC* mutants (Figure 2D). At the mating-type locus, the upregulated genes are mainly located at the *L* region, which has a low coverage of H3K9 methylation (Figure 2D). We also performed qRT-PCR to compare the silencing defects between *lsd* C-terminal mutants with the catalytic mutants (*lsd1-ao* and *lsd2-ao*). Consistent with previous findings, the catalytic mutants only showed weak silencing defects (Figure 2E). Notably, *lsd2-ΔC* mutants lost repression to a similar degree as *lsd1-ΔHMG,* suggesting that even a mild loss of function of Lsd2 has a robust effect on epigenetic silencing (Figure 2E). Furthermore, both RNA-seq and qRT-PCR results agree that *lsd1-ΔHMG* and *lsd2-ΔC* have stronger defects at the outer-centromeric and sub-telomeric regions than at the mating-type locus. Altogether, the data suggests that Lsd1 and Lsd2 function to repress transcription at the constitutive heterochromatic loci, contributing to essential epigenetic silencing in fission yeast.

### 3.4. C-Terminal Mutants of Lsd1 and Lsd2 Retain Some Enzymatic Activities

Our data suggest that the C-terminal mutations affect the additional functions of Lsd1 and Lsd2 beyond their catalytic amine oxidase-related activities. However, it was unknown to what degree the amine oxidase function is affected by C-terminal truncation. Alternatively, this result could merely reflect the differences in the functions of Lsd1 and Lsd2. To determine whether the C-terminal truncations affect the amine oxidase functions within the same protein, C-terminal truncations were introduced in strains already bearing the catalytic mutation to generate the double mutant alleles of *lsd1* and *lsd2* (*lsd1-ao-ΔHMG*, *lsd1-ao-ΔC*, and *lsd2-ao-ΔC*). While we were unable to isolate viable *lsd1-ao-ΔHMG* and *lsd2-ao-ΔC* strains following multiple transformations, we did generate viable *lsd1-ao-ΔC* strains that showed slight synthetic growth defects and temperature sensitivity compared to either single mutant allele strains (Figure 3A). Further investigation by qRT-PCR showed that *lsd1-ΔC* slightly exacerbated the silencing defect of *lsd1-ao* at the centromere (Figure 3B). These data indicate that the C-terminus-related activity of Lsd1 does not completely overlap with the amine oxidase activity of Lsd1. Deletion of the HMG domain in *lsd1-ao* (*lsd1-ao-ΔHMG*) likely causes severe defects, and would therefore be lethal. Likewise, the loss of both the amine oxidase function and the C-terminus in Lsd2 (*lsd2-ao-ΔC*) was catastrophic for the cell, consistent with the known lethality of *lsd2Δ* strains. As combining C-terminal mutation with the catalytic mutations within the same protein exacerbates the phenotype, the results indicate that the C-terminal mutants of Lsd1 and Lsd2 preserve some enzymatic activities.

### 3.5. Lsd1 and Lsd2 Serve Overlapping, but Divergent, Functions

While Lsd1 and Lsd2 are known to form a complex in *S. pombe* [34,49], the utility of Lsd1-Lsd2 complex formation is still unclear. To test whether the mutation of one Lsd protein affects the function of the other Lsd protein, we crossed each *lsd1* and *lsd2* single mutant strains, to generate all possible double mutant combinations. HMG-box domain-retaining mutants can form viable double mutant strains with the catalytic amine oxidase mutants, although they cannot be combined with one another (Figure 3C). *lsd1-ΔHMG* mutants were found to be synthetic lethal with all *lsd2* mutants, including *lsd2-ao*, further highlighting the importance of this domain. *lsd1-ΔC lsd2-ao* and *lsd1-ao lsd2-ΔC* mutants showed weak negative interactions compared to single mutant parental strains (Figure 3A), indicating that each *lsd ΔC* mutant can tolerate the loss of amine oxidase activity, but not loss of the C-terminus, in the other Lsd protein. This result further supports our assertion that the C-terminal mutants of Lsd1 and Lsd2 retain some enzymatic activities. In addition, we examined this interaction by qRT-PCR and found that the *lsd2-ΔC*-associated silencing defects at centromeric region and mating type locus are exacerbated by *lsd1-ao* (Figure 3D). This observation indicates that the two proteins work in parallel to suppress expression at these heterochromatic loci. Collectively, these data suggest that the C-terminal-related activities of Lsd1 and Lsd2 act in parallel with one another and do not fully overlap with their amine oxidase-related functions.

### 3.6. C-Terminal Mutants of Lsd1 and Lsd2 Exacerbate the Silencing Defects of ago1Δ

Since RNAi is the main pathway of assembly heterochromatin at the centromeric region [25], we next examined the relationship between Lsd1/2 mutants and RNAi. We combined *lsd1* and *lsd2* mutants with *ago1Δ*, and assessed the genetic interactions via the growth curve and dilution assays (Figure 4A,B and Appendix A). The C-terminal *lsd* mutants showed synthetic growth defects with *ago1Δ* (Figure 4A,B), but the growth rates of *lsd1-ao* and *lsd2-ao* did not seem to be affected by *ago1Δ* (Appendix A). When combined with *ago1Δ*, both *lsd1-ΔHMG* and *lsd2*-*ΔC* show stronger losses of silencing at all heterochromatic regions (Figure 4C). Surprisingly, we observed a noticeable decrease of H3K9me2 in *lsd1-ΔHMG* and *lsd2*-*ΔC* single mutant cells at all heterochromatic regions, and the levels of H3K9me2 in those mutants were further reduced when combined with *ago1Δ* (Figure 4D). This finding suggests that the C-terminal domains of Lsd proteins play an unexpected role in epigenetic silencing through a mechanism that overlaps with RNAi.

### 3.7. The Lysine Demethylase Activity of Lsd Proteins Antagonize RNAi

We next investigated whether the catalytic mutations of Lsd1 and Lsd2 affect RNAi. Unlike the C-terminal mutants of Lsd1 and Lsd2, *lsd2-ao* suppresses the silencing defect of *ago1Δ* at the centromeric region (Figure 4C), suggesting that the amine oxidase activity of Lsd2 antagonizes the function of RNAi. We did not observe the same effects at the mating-type locus or the telomeric regions, because redundant pathways are present to ensure the silencing of these regions. As weak H3K9me2 demethylase activity by Lsd1 has been previously demonstrated in *S. pombe* [39,49,62], the catalytic mutants of Lsd1 or Lsd2 may alleviate the silencing defects of *ago1Δ* through reduced demethylation of the repressive heterochromatin marker, histone H3 dimethyl-K9 (H3K9me2). This is consistent with our detection of enhanced H3K9me2 in both *lsd1-ao* and *lsd2-ao* mutants, and the cumulative effects indicate that the catalytic activities of Lsd proteins limit H3K9 methylation and heterochromatin formation (Figure 4D). Compared to *ago1Δ*, the levels of H3K9me2 increase at the centromeric regions in *lsd1-ao ago1Δ* and *lsd2-ao ago1Δ* double mutant cells (Figure 4D). In addition, the *lsd1-ao lsd2-ao* double mutants show the highest levels of H3K9me2 (Figure 4D). These results agree with the role of Lsd proteins, as H3K9 demethylases as previously implicated [39,49,62].

### 3.8. Lsd1/2 Play Roles in the Maintenance and Re-Establishment of Heterochromatin at the Mat Locus via an RNAi-Independent Mechanism(s)

To further analyze the RNAi-independent function of Lsd1/2, we introduced Lsd C-terminal mutations into cells that lack part of the *K* region at the mating-type locus, but continued to repress a proximal *ade6^+^* reporter gene (*KΔ::ade6^+^* off) (Figure 5A). The silencing of the *ade6^+^* reporter gene causes the accumulation of the red-pigmented metabolite 5-aminoimidazole ribotide (AIR), resulting in the formation of red colonies on low adenine medium [29]. Cells that partially or completely lose silencing of this reporter gene form pink sectoring colonies or white colonies, respectively [29]. Deleting the *K* region containing *cenH* (*KΔ*) results in the loss of the RNAi nucleation center within the mating-type locus [23]. In *KΔ* cells, RNAi-independent mechanisms can partially form heterochromatin and mediate gene silencing, although the function of Clr4 and Swi6 are required [75]. Once the silencing is established (*KΔ::ade6^+^* off), it can be maintained through mitotic and meiotic cell division [75]. After we combined *lsd1* and *lsd2* mutants with *KΔ::ade6^+^* off cells through genetic crosses, we detected heterogeneous expression of the *KΔ::ade6^+^* reporter in all *lsd1* and *lsd2* mutants, as observed by mixtures of red and white colonies (Figure 5B). To avoid the limitations of space and nutrition that could affect the size and color of colony formation, we placed single cells evenly on rich media plates using a dissection microscope, let the cells grow at 30 °C until colonies appear, and replica plated the colonies on low adenine medium (YE). We then assessed the expression of *KΔ::ade6^+^* using qRT-PCR, and we found that *lsd2-ΔC* cells showed the strongest expression, which is consistent with their formation of the lighter pink color on YE medium (Appendix A). Interestingly, although *lsd1-ao lsd2-ao* cells showed the weakest expression of *KΔ::ade6^+^*, these colonies show strong phenotypic variation, similar to *lsd1-ΔHMG* and *lsd2-ΔC* colonies (Figure 4B), indicating that the roles of Lsd1 and Lsd2 are involved in the epigenetic maintenance of heterochromatin.

In wild-type cells, *KΔ::ade6^+^* off (red colonies) can switch to *KΔ::ade6^+^*on, forming white colonies on YE, indicating the loss of heterochromatin at the *mat* locus. However, through cell division, white colonies that have lost the silencing of *KΔ::ade6^+^* can also re-establish silencing on this reporter gene, forming red colonies after generations of growth. To investigate how efficiently the switched white cells can re-establish heterochromatin at *KΔ::ade6^+^*, we transferred individual white cells onto rich media, using a dissection microscope. After the colonies start to appear on rich medium plates, we replica-plated colonies on low adenine medium. After propagation, about 28.5% of initially white wild-type cells formed red colonies, indicating the efficient re-establishment of heterochromatin at the silent *mat* region in an RNAi-independent manner (Figure 5C). We observed a reduction in H3K9me2 levels between red and white colonies in wild-type backgrounds by qChIP and qRT-PCR (Figure 5D,E). In *lsd1-ao lsd2-ao* cells, only about 6.1% of initially white colonies switched to mixed colonies, suggesting a defect in re-establishment of heterochromatin. Interestingly, almost 100% of *lsd1-ao lsd2-ao* red cells gave rise to variegated colonies, indicating that the catalytic activities of Lsd1 and Lsd2 proteins are important for the maintenance of heterochromatin at the *mat* locus using an RNAi-independent mechanism (Figure 5C). Similar to *lsd1-ao lsd2-ao* white cells, about 3.4% *lsd1-ΔHMG* or 5.9% *lsd2-ΔC* white cells can form pigmented colonies, suggesting a re-establishment deficiency. Our results support that these proteins play important roles in the establishment and maintenance of epigenetic silencing at the mating-type locus in an RNAi-independent manner.

### 3.9. Loss of Epe1 Suppresses the Silencing Defects of Lsd1 and Lsd2 Mutants

In addition to the two Lsd proteins, Epe1, a JmjC domain containing protein, is an anti-silencing factor predicted to function as a histone H3K9 demethylase based on sequence analysis [76,77,78,79,80], although the activity has not yet been formally confirmed. Deletion of Epe1 suppresses the silencing defects caused by the loss of most heterochromatic factors including HDACs and RNAi [76,77,78,79,80]. Epe1 also promotes histone turnover in heterochromatin and acts to prevent spreading of heterochromatin beyond its boundaries [37,40]. Since Lsd proteins are histone demethylases and *lsd1Δ* was reported to cause spreading beyond heterochromatin boundaries, [39,49] we wondered whether Lsd1 or Lsd2 is functionally redundant to Epe1. We therefore combined *lsd* mutant alleles with *epe1Δ* and assessed their genetic interactions. Dilution assays indicate that *lsd1-ΔHMG* and *lsd2-ΔC* show very weak positive interactions with *epe1Δ* in growth at the stress temperature (37 °C) (Figure 6A). qRT-PCR analysis shows that *epe1Δ* partially suppresses the silencing defects of *lsd1-ΔHMG* and *lsd2-ΔC* (Figure 6B). H3K9me2 levels are also increased in *epe1Δ lsd1-ΔHMG* and *epe1Δ lsd2-ΔC* double mutants compared to *lsd1-ΔHMG* and *lsd2-ΔC* single mutants (Figure 6C). These data indicate that the anti-silencing mechanism of Epe1 antagonizes the silencing defects of Lsd1/2, similarly to its effect on the other silencing factors [37,40,41,42,43,44,76,77,78,79,80].

### 3.10. Lsd1 and Lsd2 Coordinate with Clr3 and Sir2, but Not with Clr6, in Transcriptional Repression at Heterochromatic Regions

While human LSD1 interacts with HDAC complexes to mediate repression [53,54,55,56], *S. pombe* Lsd1 or Lsd2 does not appear to physically associate with HDACs [34,39,49]. Nonetheless, the phenotype of *lsd1Δ* mutant is similar to *clr6-1*, as described previously [34,62]. Both *lsd1Δ* and *clr6-1* display decreased growth, protrusions, and multinucleate phenotypes. In addition, a significant overlap has been found between genes repressed by Lsd1 and those repressed by Clr6 [34,62]. These results suggest that Lsd1 and Clr6 may work synergistically in the same pathway. To explore whether the repressive functions of Lsd1 and Lsd2 are connected to HDAC-mediated repression, we combined *lsd* C-terminal mutants with mutants of each of the three HDACs: *clr6-1, clr3Δ*, and *sir2Δ*, and assessed their genetic interactions by dilution assay (Figure 7A–C). Interestingly, *sir2Δ* rescues the growth defects and heat sensitivity of *lsd1-ΔHMG* and *lsd2-ΔC*, indicating that Sir2, Lsd1, and Lsd2 are epistatic in cell growth and may work in the same pathway in controlling gene expression at euchromatic loci (Figure 7C).

At all constitutive heterochromatic domains, *lsd* C-terminal mutants showed strong cumulative genetic interactions with *clr3Δ* and *sir2Δ*, indicating that they play overlapping functions in epigenetic silencing. In particular, *lsd2-ΔC sir2Δ* double mutants show the strongest silencing defects at centromeric repeat regions and sub-telomeric regions, while *lsd1-ΔHMG clr3Δ* double mutants have the most robust effect at the mating-type locus (Figure 7D). As expected, *clr6-1* shows no additive interactions with *lsd* C-terminal mutants, indicating that Lsd1 and Lsd2 may be involved in the same pathway with Clr6 in gene silencing at heterochromatic regions (Figure 7A,D). Next, we investigated whether the defective silencing in these mutants is linked to changes in heterochromatin formation. Indeed, H3K9me2 ChIPs showed the greater reduction of H3K9me2 levels in *clr3Δ lsd* or *sir2Δ lsd* double mutant cells at the constitutive heterochromatic regions, compared to single mutants (Figure 7E). Altogether, our results suggest that Lsd1 and Lsd2 operate in parallel with Clr3 and Sir2, but not with Clr6, in gene silencing at heterochromatic regions.

## 4. Discussion

As the first histone demethylases revealed in *S. pombe*, much attention has been focused on the roles of Lsd1 and Lsd2 in antagonizing histone methylation. However, emerging views strongly indicated that their essential functions rely on their non-enzymatic activities based on the observation that cells carrying both catalytically inactive Lsd proteins are viable and do not show significant growth defects. To provide insight into the complex roles of Lsd proteins beyond their well-known histone demethylation activities, we generated and investigated new mutants of *lsd1* and *lsd2* with intermediate phenotypes and examined their functional interactions with each other. We demonstrated that the N-terminal domain and the C-terminal HMG-box domain of Lsd2 are essential for viability, while the HMG domain of Lsd1 is important, but not essential, for growth and morphology, suggesting divergent roles in *S. pombe* for Lsd1 and Lsd2. Notably, the C-terminus of Lsd1 was shown to perform functions that do not completely overlap with the amine oxidase-related demethylation activities. Instead, we found that Lsd1 and Lsd2 repress heterochromatic transcripts, probably through both RNAi-dependent and -independent manners. Our data also suggest that Lsd1 and Lsd2 regulate each other through a previously undescribed mechanism.

### 4.1. The N-Terminus of Lsd2

We identified a region spanning 13 amino acids (SFYSMNTSENDPD) near the N-terminus of Lsd2, which is required for viability. Our results indicate that the nuclear localization of Lsd2 is affected by the loss of this region, suggesting that it contains critical residues that mediate the nuclear localization of Lsd2. While human LSD1 contains a predicted nuclear localization signal (NLS) _112_RRKRAK_117_ [81], we could not find the similar NLS sequence in *S. pombe* Lsd1. We also did not find the similar 13 amino acid NLS segment in human LSD2. In addition, human LSD2 has an N-terminal zinc finger domain that seems to be required for histone demethylase activity and may mediate other molecular interactions [56], but this domain is not found in *S. pombe* Lsd2. The fact that *S. pombe* Lsd proteins share about 40% sequence similarity to that of human homologs [56], while Lsd proteins are completely missing in budding yeast, suggests that Lsd proteins are essential in organisms that have H3K9 methylation. This study provides the first evidence of the importance of the N-terminus of Lsd2 in *S. pombe*.

### 4.2. Mutual Regulation of Lsd1 and Lsd2

Our data implicate that Lsd1 and Lsd2 regulate each other through an unknown mechanism, unlikely to be related to histone demethylation. While functions of Lsd proteins beyond histone demethylation have not been well-demonstrated in *S. pombe*, non-histone substrates have been described for human LSD1 and LSD2. For instance, LSD1 demethylates p53 to inhibit apoptosis [82], targets IFITM3 (interferon-inducible transmembrane family protein 3) to restrict influenza A virus infection in response to IFNα [83], and has been shown to activate HIV transcription through the demethylation of the HIV Tat protein [84]. LSD2 has been found to target non-histone substrates as well, promoting the degradation of OGT (O-GlcNAc transferase) through unexpected E3 ubiquitin ligase activity, unrelated to its amine oxidase function [85]. Alternative mechanisms have also been suggested for LSD1, which promotes prostate cancer proliferation through an amine oxidase-independent mechanism [86]. In *S. pombe*, Lsd1 and Lsd2 may regulate one another through direct demethylation. However, up to date, no non-histone substrates have been identified for these enzymes. Alternatively, it is possible that the cross-regulation of Lsd1 and Lsd2 is not mediated by demethylation, but by demethylation-independent functions related to their C-terminal domains. Either way, the notion that Lsd1 and Lsd2 regulate the activities of one another within the same complex is intriguing and will be further investigated in our future studies.

### 4.3. The Catalytic Activities of Lsd Proteins Antagonize RNAi

RNAi is the primary pathway of heterochromatin assembly at the centromeric regions, but is dispensable for the maintenance of heterochromatin at the *mat* locus and sub-telomeres [8]. The simultaneous loss of Lsd1 and Lsd2 catalytic activities slightly increases RNA levels from the repetitive regions (Figure 4C), suggesting that other H3K9 demethylases, such as Lid2 or Epe1, play major roles in the dynamic assembly of heterochromatin at these regions [45,76]. In the absence of RNAi, the impaired amine oxidase activities of Lsd1 and Lsd2 alleviate the silencing defects, resulting in the reduction of repeat transcript levels and the elevation of H3K9me2 levels at the main heterochromatic regions. Previous studies showed that both Lsd1 and Lsd2 are enriched at very low levels at specific heterochromatic regions, compared to their levels at euchromatic regions [34,49]. It is possible that, in the context of elevated transcription at centromeric repeat regions in cells lacking RNAi machinery, the enrichment of Lsd proteins is enhanced, which facilitates the demethylation of H3K9 and further the transcription of the repeats. It is also possible that catalytically inactivated Lsd proteins increase their association with chromatin and physically occlude the recruitment of another demethylase [87].

### 4.4. The Role of Lsd Proteins in Maintenance and Re-Establishment of Epigenetic Silencing

Heterochromatin assembly requires sequential steps [8,29]: it is nucleated at designated genomic loci, spread to surrounding regions [88], and its structure is maintained during DNA replication and propagated through multiple cell divisions [19,77]. Factors that are involved in heterochromatin formation often participate at specific step(s) rather than throughout the process [8]. In particular, the RNAi silencing pathway is required for the nucleation of heterochromatin formation [88]. Once silencing is established, RNAi machinery is dispensable [89]; the heterochromatic state can be maintained without RNAi. At the mating-type locus, RNAi machinery is required to nucleate heterochromatin at the *cenH* region [46], but is dispensable for its maintenance [23]. In addition to RNAi machinery, multiple mechanisms are present at the mating type locus to ensure the efficient establishment of heterochromatin [27,28,29]. Without the *cenH* region, RNAi-independent pathways can form heterochromatin, inefficiently, at the *K* region [75,89]. This finding is based on the studies using *KΔ::ura4^+^* off allele [75,89]. In addition to *ura4^+^*, *ade6^+^* gene is a widely used reporter in *S. pombe* for RNAi-dependent and independent heterochromatin formation at centromeric regions and the mating-type locus [29,77,90,91]. When the K region is replaced by an *ade6^+^* reporter gene, the reporter gene can be silenced by heterochromatin (*KΔ::ade6^+^* off) [29]. Although the silenced status of this allele can be maintained through multiple generations, a low percentage of *KΔ::ade6^+^*off cells lose silencing and switch to *KΔ::ade6^+^*on status, accompanied by the loss of heterochromatin (Figure 5C, WT, R to W). Gene silencing and heterochromatin can also be re-established during cell division (Figure 5C, WT, W to R). Using this system, we found that pre-established *KΔ::ade6^+^*off alleles are unstable in all Lsd mutants (Figure 5). Interestingly, although cells that lack catalytic activates of both Lsd1 and Lsd2 show fairly normal growth, they form phenotypically variable colonies, indicating the unstable inheritance of the *KΔ::ade6^+^*off allele. Once the cells lost the heterochromatin and formed white colonies, all Lsd mutants struggled to re-establish the *KΔ::ade6^+^*off state, indicating the defective assembly of heterochromatin at the *mat* locus without the *K* region. In *lsd1-ao lsd2-ao* double mutant cells, we consistently detected enhanced H3K9me2 on *KΔ::ade6^+^*off alleles, but still observed phenotypic variegation, suggesting that heterochromatin may not be the sole mechanism to explain this phenotype. In addition, peri-centromeric, but not mating-type or sub-telomeric regions, show great overlap between changes in H3K9me2 enrichment and gene expression in the C-terminal mutants of Lsd1 and Lsd2 (Figure 2D). This observation further supports that the roles of Lsd1 and Lsd2 in gene silencing are not exclusively reliant on H3K9me. Overall, our results suggest that both catalytic and non-catalytic roles of Lsd proteins are involved in heterochromatin maintenance and re-establishment.

### 4.5. The Roles of LSD Proteins beyond Their Catalytic Activities

We consistently detected cumulative silencing defects and the reduction of H3K9me2 when combining both C-terminal mutants of Lsd proteins with *ago1Δ* (Figure 4C,D). This result suggests that the functions of Lsd proteins play dominant roles in epigenetic silencing at all heterochromatic regions besides their catalytic activities. They may operate within protein complexes formed with additional factors that exhibit coregulatory or scaffolding functions. In mammals, such interactions have been shown to modulate catalytic activities, substrate specificities, and/or localizations of Lsd proteins that link to various biological processes [56]. For example, LSD1 interacts with different transcriptional repressors, including Co-REST and Co-REST like proteins, which enforces repressed chromatin status and gene silencing [50,51,52]. LSD1 also interacts with the NuRD complex, which displays nucleosome remodeling and HDAC activity, and is important for a wide variety of cellular processes [53,54,55,56], In *S*. *pombe*, in addition to H3K9 methylation, the deacetylation of histones by histone deacetylases (HDACs) is essential for gene repression and heterochromatin assembly [31,46,92]. Although direct physical interactions with HDACs were not identified [34,49], *S. pombe* Lsd proteins cooperate with HDACs.

At heterochromatic regions, the C-terminal mutants of Lsd proteins show strong additive silencing defects with Clr3 (Class II) and Sir2 (Sirtuin family), but not with Clr6 (Class I) HDACs. This finding suggests that Lsd proteins coordinate with Clr3 and Sir2 in heterochromatin assembly and repression of the repeat transcripts. In a genome-wide study of HDAC functions, Clr6 was found to be the main HDAC involved in promoter-localized repression, while Sir2 and Clr3 target many loci for repression, mainly in heterochromatic regions [31,33]. As significant overlap has been described between genes repressed by Lsd1 and those repressed by Clr6, our observation that the combination of *clr6-1* and the C-terminal mutants of Lsd proteins have a weak or no genetic interaction in cell growth (Figure 7), is consistent with a model in which Lsd proteins act in concert with Clr6. We also found that *sir2Δ* partially alleviates the growth defects of *lsd* C-terminal mutants. This positive genetic interaction suggests that the roles of Lsd proteins may act in the same pathway with Sir2 at euchromatic loci. However, results from dilution assays provide a whole-cell summary of the genetic interaction and may not reflect genetic interactions with respect to a particular function (e.g., heterochromatic silencing; Figure 7). Considering the complicated and multifaceted regulation of gene expression by Lsd proteins, the antagonistic interaction between Lsd1/2 with Sir2 reflects the additional functions of Lsd1/2, beyond epigenetic silencing.

## 5. Conclusions

Here, we explore the catalytic-dependent and -independent roles of conserved lysine demethylases Lsd1 and Lsd2 in epigenetic silencing of constitutive heterochromatin in *S. pombe*. We reveal several novel findings: (1) an N-terminal peptide is essential for the nuclear localization of Lsd2; (2) Lsd1 and Lsd2 regulate each other; (3) the catalytic roles of Lsd1 and Lsd2 antagonize RNAi in heterochromatin formation; (4) Lsd1 and Lsd2 play roles in the maintenance and establishment of heterochromatin; and (5) Lsd1 and Lsd2 cooperate with Class II and Sirtuin family HDACs in constitutive heterochromatic silencing. A summary table showing all the genetic interactions between the Lsd1/2 mutants and silencing factors that were investigated in this study is available in Appendix A.

## Figures and Tables

**Figure 1 cells-09-00955-f001:**
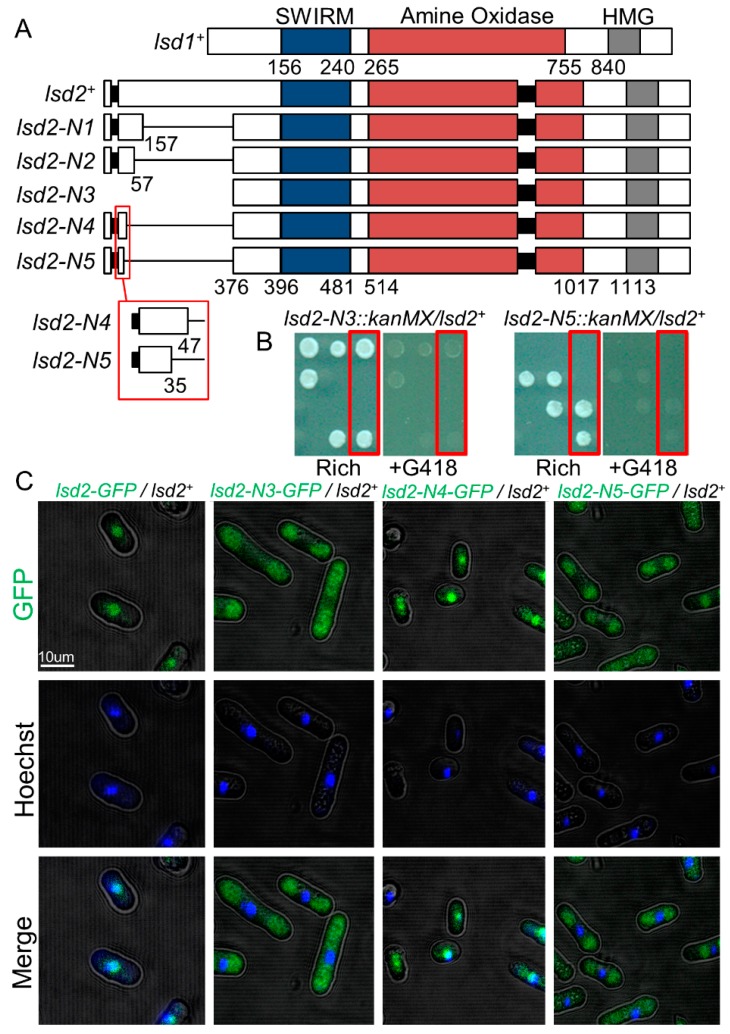
Nuclear localization of Lsd2 is required for viability. (**A**) Schematic representation of N-terminal mutants of Lsd2. Numbers indicate the position in the amino acid sequence of Lsd2 and Lsd1. (**B**) Lsd2 N-terminal mutants *lsd2-N3* and *lsd2-N5* are inviable. Heterozygous diploids containing one copy of the mutant allele (linked with resistance to the antibiotic G418) and one copy of the wild-type (WT) allele were sporulated and tetrads of spores were arranged in columns. Red box: sample of a tetrad that shows selective growth on rich media with or without G418. (**C**) Visualization of GFP-tagged full-length or N-terminal truncated Lsd2 proteins in viable heterozygous diploid strains by confocal fluorescence microscopy. Indicated strains were also stained with Hoechst to show nuclear localization.

**Figure 2 cells-09-00955-f002:**
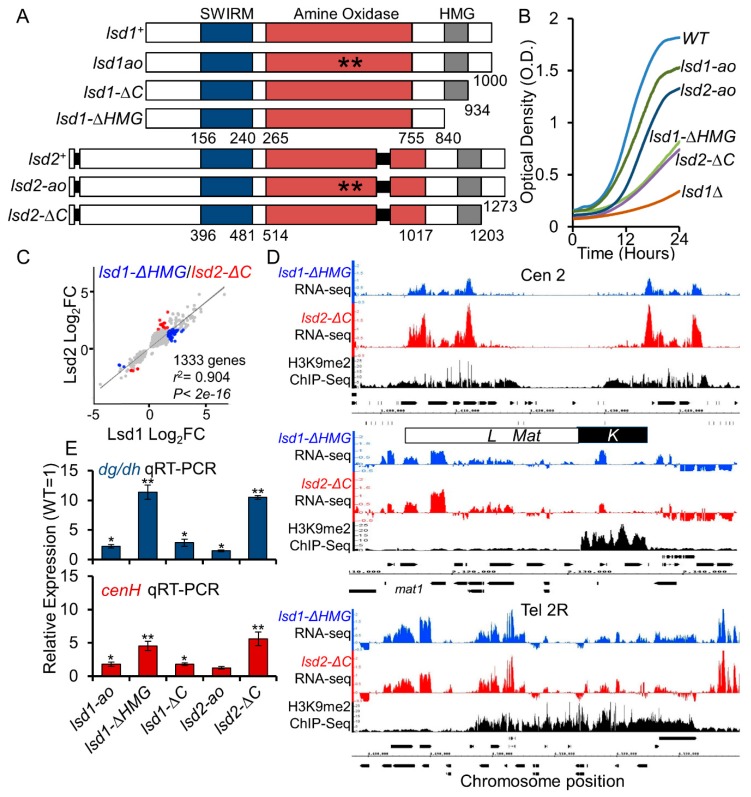
The C-terminal domains of Lsd1 and Lsd2 are vital for proper growth and genome-wide gene expression. (**A**) Schematic representations of WT Lsd1 and Lsd2 compared to catalytically inactive and C-terminal truncated mutants. (**B**) Growth curves generated by a plate reader show the growth rates of indicated WT, *lsd1*, and *lsd2* mutants over a 24 h period. (**C**) Comparison of DEGs between *lsd1-ΔHMG* and *lsd2-ΔC* using linear regression analysis. (**D**) Loss of silencing at peri-centromeric region, mating type locus, and sub-telomeric region in *lsd1-ΔHMG* and *lsd2-ΔC* compared to WT. Cen 2: Centromere II. *Mat*: Mating-type locus. Tel2R: right telomere II. Normalized RNA-seq reads are aligned with H3K9me2 enrichment as measured by ChIP, which indicate heterochromatic regions. Data are plotted along with chromosome position. (**E**) qRT-PCR analysis of the silenced *dg/dh* repeats in the peri-centromeric region and *cenH* (*mat* locus). * *p* ≤ 0.05 and ** *p* ≤ 0.01 as determined by student’s *t* test comparing the indicated samples with WT. Error bars represent *s.e.m.*

**Figure 3 cells-09-00955-f003:**
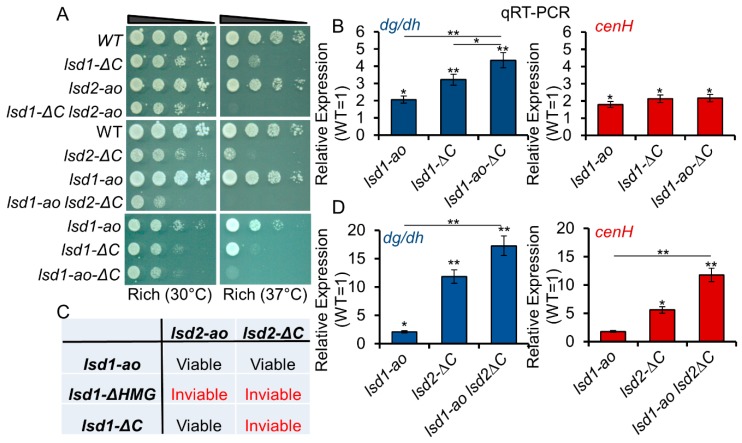
Lsd1 and Lsd2 serve overlapping, but divergent, functions. (**A**) Serial dilutions demonstrate the synthetic growth defects and heat sensitivities of *lsd1 lsd2* double mutants. (**B**) qRT-PCR analysis of peri-centromeric (*dg/dh*) and mating-type (*cenH*) regions show the genetic interactions between *lsd1-ao* and *lsd1-∆C* and the double mutant allele *lsd1-ao-∆C.* (**C**) Summary of the genetic interactions between *lsd1* and *lsd2* mutants. (**D**) qRT-PCR analysis of the silenced *dg/dh* repeats in the peri-centromeric region and *cenH* (mating-type locus) show the genetic interactions between *lsd2-ΔC* and *lsd1-ao.* * *p* ≤ 0.05 and ** *p* ≤ 0.01 as determined by student’s *t* test comparing the indicated samples with WT values. Significance between single mutants and double mutants is indicated by horizontal lines linking the two samples. Error bars represent *s.e.m.*

**Figure 4 cells-09-00955-f004:**
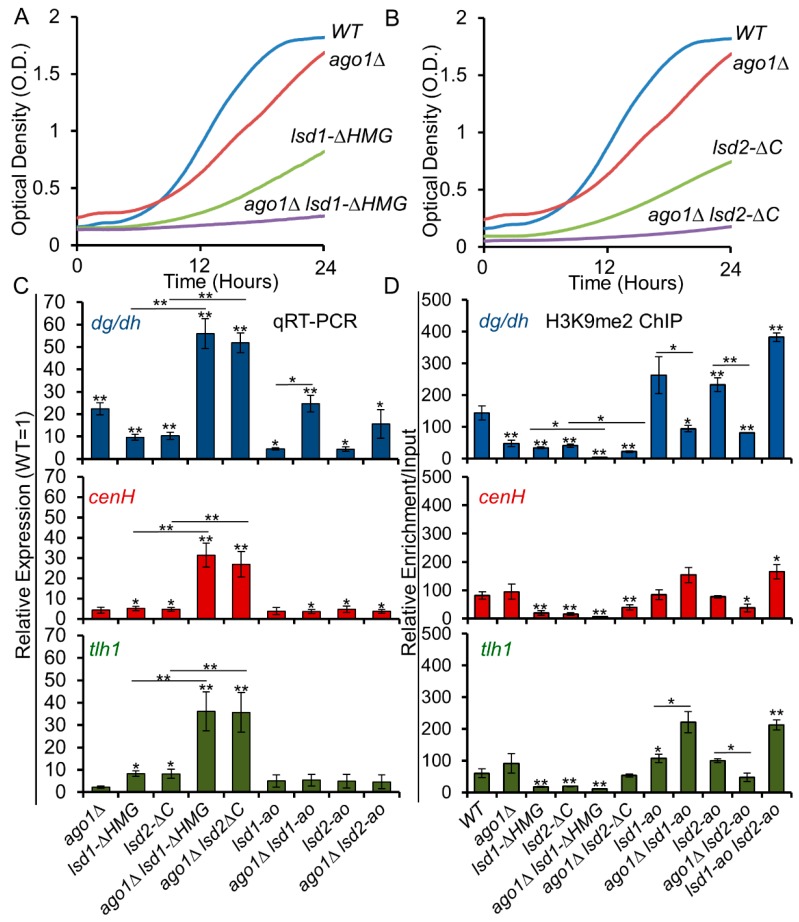
The silencing defects of RNAi mutant *ago1Δ* are suppressed by Lsd1/2 amine oxidase mutants, but are exacerbated by Lsd1/2 C-terminal mutants. (**A**,**B**) Growth curves generated by a plate reader over a 24 h period show the genetic interactions between *lsd1* and *lsd2* C-terminal truncated strains combined with *ago1Δ*. (C-D) qRT-PCR (**C**) and qChIP (**D**) analyses of silent *dg/dh* repeats (centromeric)*, cenH* (mating-type locus), and *tlh1* (telomeric) regions demonstrate the genetic interactions between *ago1Δ* and *lsd* mutants. * *p* ≤ 0.05 and ** *p* ≤ 0.01 as determined by student’s *t* test comparing the indicated samples with WT for qPCR or qChIP and by horizontal lines linking the two samples. Error bars represent *s.e.m.*

**Figure 5 cells-09-00955-f005:**
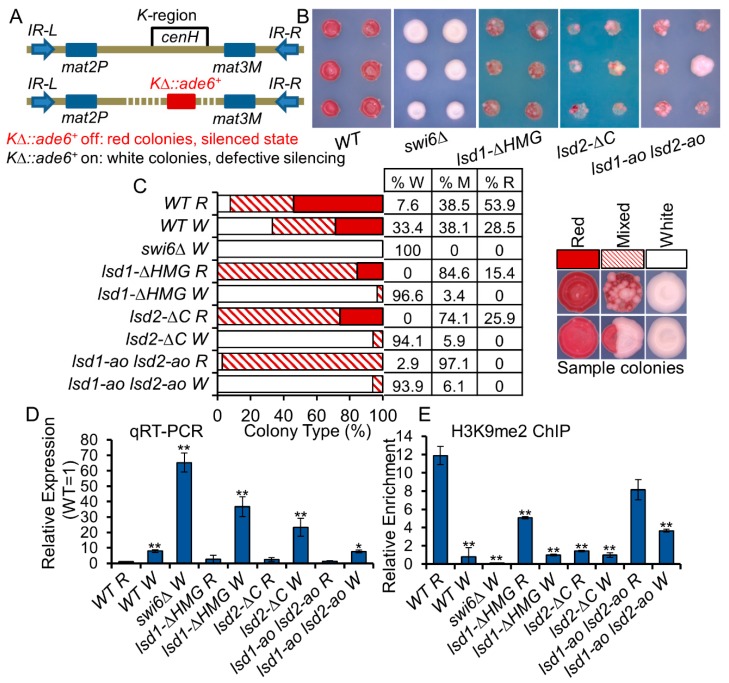
Lsd1 and Lsd2 mutants affect epigenetic maintenance and re-establishment of heterochromatin. (**A**) An *ade6^+^* reporter gene replaced the K region (*K∆::ade6^+^*). Epigenetic mechanism(s) can repress *ade6^+^* expression (*K∆::ade6^+^* off, red colonies); loss of the epigenetic silencing causes expression of *ade6^+^* (*K∆::ade6^+^* on, white colonies). (**B**) The *K∆::ade6^+^* off allele was introduced into cells with indicated genotypes via genetic crosses. Samples of color and shape of colonies formed by individual cells on low adenine medium are shown. Colonies were replica-plated from rich medium (YEA) to low adenine medium (YE). (**C**) Cells that formed red or white colonies from (B) with indicated genotypes were dissected and grown on rich medium without selection, then were transferred to low adenine medium. Samples of red (R), white (W), and color-mixed (M) colonies on YE medium are shown on the right. Bar graph to the left shows the percentage of colonies that are red, white, or color-mixed. The percentages are listed in the table for each of the indicated genotypes. (**D**,**E**) *ade6^+^* expression in red or white cells collected in (C) were investigated by qRT-PCR (D), and relative H3K9me2 enrichment at repeat regions (versus input) (E) were compared in red or white colonies with indicated genotypes. * *p* ≤ 0.05 and ** *p* ≤ 0.01 as determined by student’s *t* test comparing the indicated samples with WT R for qPCR or qChIP. Error bars represent *s.e.m.*

**Figure 6 cells-09-00955-f006:**
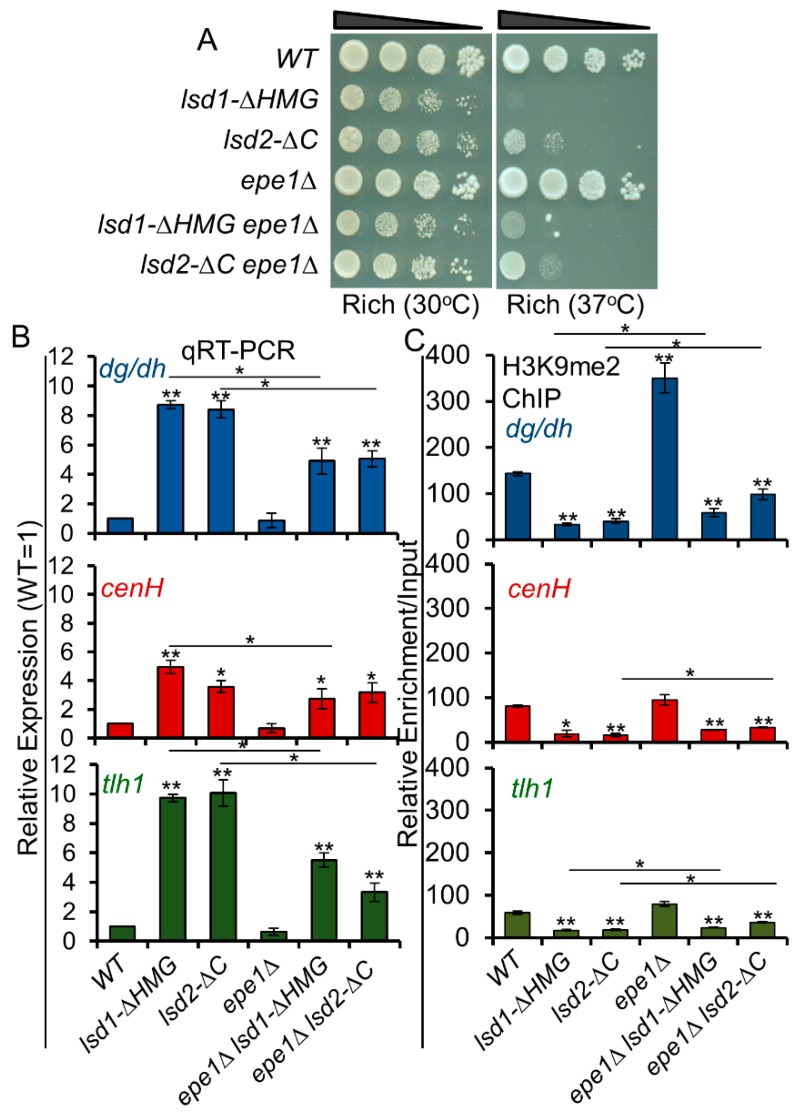
Epe1 antagonizes Lsd1 and Lsd2 functions. (**A**) Serial dilutions show weak synthetic growth defects when *epe1Δ* is combined with *lsd1-ΔHMG* or *lsd2-ΔC* at 37 °C. (**B**) qRT-PCR analysis demonstrates the suppressive effects of *epe1Δ* on *lsd* mutant silencing defects at all heterochromatic regions. (**C**) The alleviation of the silencing defects by *epe1Δ* on *lsd* mutants are correlated with the alterations of H3K9me2 detected by ChIP. * *p* ≤ 0.05 and ** *p* ≤ 0.01 as determined by student’s *t* test comparing the indicated samples with WT for qPCR or qChIP and by horizontal lines linking the two samples. Error bars represent *s.e.m.*

**Figure 7 cells-09-00955-f007:**
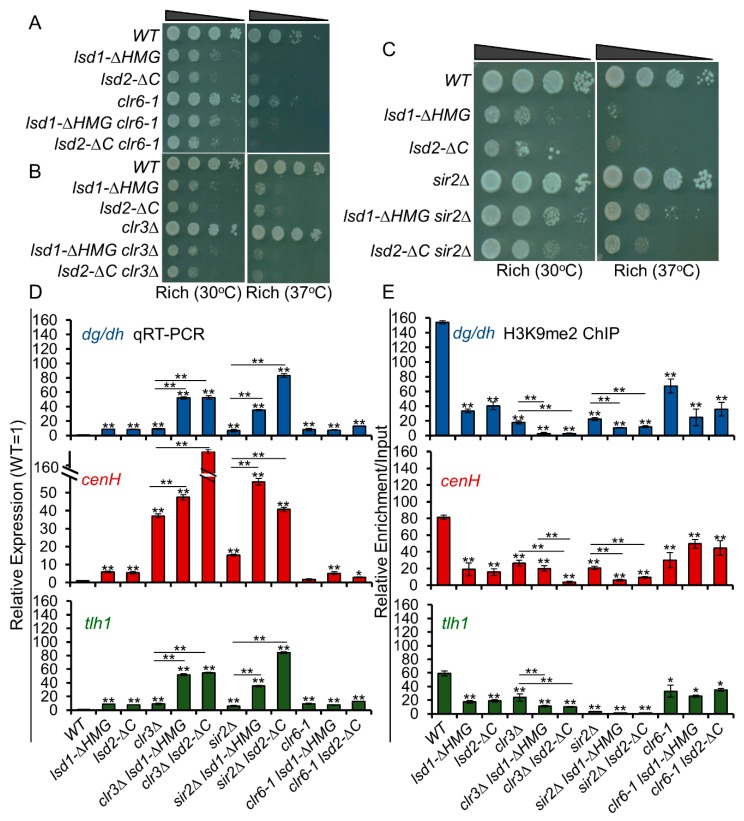
Lsd1 and Lsd2 mediate overlapping functions with Clr3 and Sir2, but not with Clr6, at heterochromatic regions. (**A**–**C**) Serial dilutions show genetic interactions with respect to growth and heat sensitivity between *lsd* C-terminal mutants and mutants of histone deacetylases *clr6-1* (**A**), *clr3Δ* (**B**), and *sir2Δ* (**C**). (**D**) qRT-PCR analysis shows cumulative silencing defects of *Lsd* mutants combined with HDAC mutants at the centromeric (*dg/dh*), mating-type locus (*cenH*), and telomeric (*tlh1*) heterochromatin regions. (**E**) H3K9me2 ChIP analysis demonstrates the alterations of heterochromatin in WT, single, and double mutant cells with the indicated genotypes. * *p* ≤ 0.05 and ** *p* ≤ 0.01 as determined by student’s *t* test comparing the indicated samples with WT for qPCR or qChIP and by horizontal lines linking the two samples. Error bars represent *s.e.m.*

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
