# Peer review of "The Catalytic-Dependent and -Independent Roles of Lsd1 and Lsd2 Lysine Demethylases in Heterochromatin Formation in Schizosaccharomyces pombe"

_cells, 2020, doi:10.3390/cells9040955_

Round 1

Reviewer 1 Report

See attached file

Author Response

Dear reviewer,

We appreciate your kind help in reviewing our manuscript.  In response to your insightful comments by the reviewers, we have revised the paper by performing additional experiments and modifying the figures and text.  We are grateful for the improvements and thank you for your useful comments.

Below is our detailed response to specific comments by the reviewers, which are indicated in italics:

Paper strengths:

-very good structure/function characterisation of proteins with high importance but lacking significant sequence homology with well-characterised human homologues. This paper contributes a lot to the field of both histone demethylase function, as well as the distinction between RNAi dependent and independent silencing in S. pombe.

- The paper for the most part uses genetic approaches to dissect the complex function of these histone demethylases and provides many novel insights into the different molecular pathways in which these proteins operate. The generation of the unique and viable C-terminal deletion mutants of Lsd1 and especially the essential gene Lsd2 will be a great resource for the S. pombe field going forward.

- The identification of a non-canonical NLS in the N-terminus of Lsd2 is a very interesting find.

We are grateful for the favorable statement from this reviewer.

Overall comment:

This reviewer appreciates all the work that went into the data and how they represents a great many experiments, however authors needs to be more discernible in the data they choose to present in the main paper and what their take home messages are (see below)…. there is a lot of data here, and needs to be condensed better into some definitive conclusions. Additionally a summary table of the results with all the genetic interactions between the Lsd1/2 C-terminal deletions which were investigated, would be very useful.

We agree with this comment. We have inserted a summary table of the results with all genetic interactions between Lsd1/2 C-terminal mutants with the heterochromatic factors investigated in this study. Please find the table in Supplemental Figure 10.

Paper Caveat:

-one caveat to the paper is that the authors did not test whether the C-terminal mutations of Lsd1 and Lsd2 impact their enzymatic activity. Clearly they display more phenotypes than the respective catalytically dead mutations, however it can’t be ruled out that they have no effect on demethylase activity. Figure 7 which combines these two alleles in Lsd1 at least, and shows synergistic effects of both mutations on cell growth would suggest that the Lsd1ΔC mutation retains some enzymatic activity….this figure should be placed earlier in the manuscript and/or demethylase assays undertaken for the C-terminal mutations.

This is a good point. Although we did not perform the in vitro demethylase assay, our genetic data presented in Figure 7 (original submission) indicate that the C-terminal mutants of Lsd1 and Lsd2 retain some enzymatic activities, as combining C-terminal mutation with the catalytic mutations within the same protein exacerbates the phenotype. As the reviewer suggested, we moved this figure to the new Figure 3.

Introduction: Very good. However, would include more information on Clr6.

We have added an extra sentence about Clr6 in the introduction.

“A third HDAC in S. pombe, Clr6, has been shown to be primarily involved in repression of euchromatic loci via deacetylation of promoter regions [31, 32], although it also controls transcription of repetitive regions including dg/dh repeats and retrotransposons [32, 34].

Materials and Methods: very scant. Many assays/protocols not described either at all or in insufficient detail. These include:

  1. Strain construction - much of the information is absent, and there is no comprehensive list of strains used in the study. There shouldn’t be details in the legends of figures that aren’t discussed in this section eg. the otr::ura4+ reporter or the FTP-tagged version of Lsd1/2.

We have included a list of strains used in this study as the Supplemental Table S1.

  1. Primer sequences are missing for both strain construction and RT-qPCR quantification.

We have also included a list of oligoes used in this manuscript as the Supplemental Table S2.

  1. Heterochromatin formation/maintenance assay using ade6 reporter (see below).

We added a new 2.7 section for a detailed procedure of performing this assay. 

  1. Calcofluor white staining

We inserted the method of the Calcofluor stain and Hoechst stain in section 2.6. “Calcofluor white staining was performed as previously established [62]. Hoechst stain for GFP live-imaging was followed as shown formerly [63]”.

  1. Spot assays - no idea how long these plates were incubated for - did it differ between experiments?

We added the requested information in section 2.2. “All cultures were grown at 30°C (or at 37° where indicated) for 2-3 days until single colonies appear in the most diluted spot.”

Results:

The results section needs a lot of work, and potentially reordering of figures (see below). Not all data are discussed in the order with which they are presented in figures, which make the reading somewhat difficult, and some figure panels are never mentioned until the discussion.

We appreciate the reviewer’s comments about this section. We have re-ordered the Figures and discussed most if not all results in the order, please see below.

Figure 1

A

- include Lsd1p for comparison

- deletions not clear - number the residues at the start and end of deletion for each construct eg lsd2-N1 number 157 and 376

- deletion schematics don't match up with the actual deletion alleles, please adjust

We have revised the Figure 1. We included Lsd1 for comparison, added the number residues at the start and end of deletion for each construct, and adjusted the schematics to match up with the actual deletion alleles.

B

- uninformative.

- If there are growth defects between the viable deletion alleles - can’t tell. Either quantify this growth difference as change in doubling time, or remove

- for the lethal alleles, lack of growth is uninformative. Show the tetrads here instead of in the supplementary information.

We moved Figure 1B to supplemental Figure 1 and moved the tetrad assay from supplemental Figure to Figure 1B.

Figure 2

- A : as with figure 1A. Number the amino acids that flank the deletions

We added the numbers as the reviewer suggested.

- B : again tetrad analysis is more informative than the spot assays. the growth defects are clearer in the tetrads. Heat sensitivities can go in the supplementary info.

Good suggestion. We would like to compare the growth rate between wild-type and lsd C-terminal and catalytic mutants. As we don’t have the tetrad assays for the catalytic mutants, we replaced the dilution assay with the growth curve. Please see the new Figure 2B.

- C,D and E too small to read

We now made them bigger and more readable.

- volcano plots not that informative - maybe supplementary info?

We moved volcano plots to supplemental Figure S8.

- D : excellent comparison, very interesting and conclusive

Thanks! Now it is C.

- how does your lsd1-ΔHMG RNAseq data match up with Lsd1 ChiP from :Lan F, et al. Mol Cell. 2007; 26(1):89-101. PubMed PMID: 17434129?

We did not compare our RNA-seq data with the ChIP data from Lan F, et al. Mol Cell. 2007; 26(1):89-101. PubMed PMID: 17434129 in a genome-wide manner. However, we tested a few top targets identified from the previous ChIP assay (Lan et al.). The changes of gene expression of the targeted genes are correlated with the loss of Lsd1 enrichment at their promoter regions. On example was shown in Figure 8. However, we decided to remove this figure from the manuscript as we agree with the reviewer that we have too much data and causing more unanswered questions (see below).

E : include mat locus here too. Is there a significant enrichment for DEG in heterochromatic regions? Not great overlap with the H3K9me ChIP and unregulated genes in the mutants - except for at Cen2. This needs discussing.

We appreciate this important point. We have included the mat locus here in Figure 2D. We also discussed this result in section 4.3,  “In addition, only peri-centromeric, but not mating-type or sub-telomeric regions show great overlap between changes in H3K9me2 enrichment and expression in the C-terminal mutants of Lsd1 and Lsd2 (Figure 2D). This observation further supports that the roles of Lsd1 and Lsd2 in gene silencing are not exclusively reliant on H3K9me.”

Supplementary figure 2 - useless, and complete conjecture. If you want to say suppressors are likely, prove it, or remove the figure.

We have removed it.

Figure 3

- A : Only convincing synthetic interaction between mutants is with lsd2ΔC and ago1Δ (final strain on the spot assay) and no others. If there is a discernible difference between ago1Δ and the other C-terminal deletions, it is not obvious in this assay and should probably be quantified by measuring strain doubling times.

As the reviewer suggested, we measured strain doubling times and replaced Figure 3A with two growth curves.

- C : the text reads “At the centromeric region, lsd2-ao suppresses the silencing defect of ago1Δ (Figure 3C)” however the significant (*) result highlighted in the figure is between lsd1-a0 and ago1Δ?

Thank you, we have fixed this mistake.

- C. can all the qRT-PCR graphs have the same scale on the y-axis. - D. can all the ChIP graphs have the same scale on the y-axis. repeat this for all figures - the data are very misleading when the scales are different between graphs.

We changed all qRT-PCR and ChIP graphs to have the same scale on the Y-axis

The text and figures do not match up well in this part of the paper - suggestion: talk about the C-terminal deletions + ago1Δ, before the catalytically dead mutants, as this is the major finding from these experiments and is what the reader is drawn to when looking at the data.

We are grateful for this comment, and we have made the changes as the reviewer suggested. Please see the new 3.6 and 3.7 sections.

-D: very interesting data

Figure 4

- Great reporter to use to assess RNAi independent silencing, and data look very promising but are not analysed correctly. (see below)

- C: as this experiment presumably was done with a mixed population of cells (different phenotypes) it isn’t very informative, as it reflects a population average of expression. The clean and informative data is in figure 4E when the R and W colonies are separate out. I would advise putting 4C in the supplementary data.

The reviewer is correct. We moved 4C to supplementary Figure S5.

- D: very informative data and definitely shows that both heterochromatin establishment and maintenance is affected by the lsd mutants. Would like to see (a) more information provided in the materials and methods on how this assay was done and the number of cells/colonies counted for each.

We have added detailed information in the materials and methods. Please see section 2.7.

 (b) these data converted to a table of switching rates (phenotype change/cell division) as well as a quantification of the heritability of the phenotype (maintenance). Also it is very easy to miss the ‘3%’ variegated colonies for the lsd mutants..need to make this more obvious.

This is a good suggestion, we further analyzed the data and the detailed information is now included as Figure 5C. We also made changes in the text accordingly. Please see section 3.8.

- D - remove ‘Legend’ under the colony colour examples.

Legend has been removed.

- F: why is there no increase in H3K9me2 in lsd2-ΔC red colony even though ade6 is silenced?

We appreciate this question. We do see slight increases of H3K9me2 in lsd2-ΔC red colonies, although not to the same degree as lsd1-ΔHMG red colonies. This finding suggests that heterochromatin may not be the sole mechanism to explain the silencing of ade6+. Also, in lsd1-ao lsd2-ao double mutant cells, we consistently detected enhanced H3K9me2 on the KΔ::ade6+off allele, but still observed phenotypic variegations, further supporting this point. We have discussed this result in section 4.3.

Figure 5

- A: do not agree with this statement “Dilution assays indicate that lsd1-ΔHMG and lsd2-ΔC show very weak positive interactions with epe1Δ in growth “. No differences in growth are evident… - B: q-RT-PCR data very convincing

We have changed this to “Dilution assays indicate that lsd1-ΔHMG and lsd2-ΔC show very weak positive interactions with epe1Δ in growth at the stress temperature (37oC)”.

Figure 6

- A + B: only convincing genetic interaction is with sir2Δ.

We agree with the reviewer, but growth curves for these many mutants are very busy. Also, we would like to keep the dilution assays for all tested HDACs in the same figure for comparisons.

- D +E: not clear which bars are compared for the statistical analysis.. make this more obvious.

We have fixed the problem. See the new Figure 7.

Figure 7

- A, B and C would’ve liked to see these data earlier in the paper perhaps? Very crucial experiments.

We have moved this Figure to the new Figure 3 and changed the text accordingly. See section 3.4 and 3.5.

- if C and E are both RT-qPCR datasets, label the y-axis the same for both, and remove qChIP from the legend.

We insist that we keep the different scales for C and E. The difference for E is much smaller than C, but is still statistically significant. If we change them to the same scale, it looks like that there is not difference between lsd1-ΔC and lsd1-ao- ΔC, which could be misleading as well.

Figure 8

- A: explain this schematic here and not in the discussion…

- does the RNAseq data also show that sah1 is a DEG for the lsd C-terminal mutants? Does the RNAseq data indicate/suggest that there might be other reasons also?

- C and D: y-axis should be log2 with X-axis crossing at 1

- E: assay was not described in the methods, and the results are not discussed in the text aside from in the discussion.

Conclusions from these data are difficult to make given that sah1Δ does not cause silencing defects itself. Seems this experiment raises more questions than answers.

We agree with the reviewer. Since we have too much data presented in this manuscript, we have decided to remove this figure to avoid unnecessary confusion.

We believe that we have addressed all the points raised. We also hope that these changes are satisfactory.

Sincerely,

Ke Zhang

Associate Professor

Department of Biology

Wake Forest University

455 Vine Street

Winston-Salem, NC 27101

Reviewer 2 Report

LSD1 and LSD2 are highly conserved lysine-specific demethylases, and their well-characterized targets are mono- or dimethylated histone H3 (K4 or K9). Fission yeast S. pombe expresses Lsd1 and Lsd2, and previous studies described their roles in heterochromatin assembly and euchromatic gene regulation. However, the precise roles of these proteins have not clearly defined yet. In this manuscript Marayati and colleagues describe the catalytic-dependent and independent roles of Lsd1 and Lsd2 in heterochromatin formation. The author first show that Lsd2 N-terminal region is essential for viability and regulate its nuclear localization. The authors also show that the C-terminal regions of Lsd1 and Lsd2 play pivotal roles in proper growth and genome-wide gene expression. The authors further demonstrate that C-terminal deleted- and catalytically deficient mutants of Lsd proteins displays distinct effects on heterochromatin assembly when combined with the RNAi-deficient mutant. The authors then show that while the catalytically deficient mutants of Lsd proteins have a minor effect on growth, they affect the epigenetic maintenance of heterochromatin. The authors further describe the functional relationship between Lsd proteins and other histone modifying enzymes. Finally, the authors show that although Lsd proteins are essential for sah1+ expression, the silencing defects of Lsd C-terminal mutants are not mediated by the change of sah1+ expression.

Comments:

The mechanism how Lsd proteins regulate higher-order chromatin assembly and euchromatic gene expression has been an important unresolved question, and this study clearly demonstrate the catalytic activity-independent role of S. pombe Lsd proteins in heterochromatin assembly. Although previous studies have already described a minor effect of catalytically deficient mutants of Lsd proteins in growth and heterochromatin assembly, this study provide further insights into how they cooperatively function to regulate heterochromatin. The results presented are in most cases of high quality and convincingly controlled. I have the following concerns that should be addressed by the authors before publication.

Major points:

1.  S. pombe Lsd proteins stably interact with two PHD finger proteins, Phf1 and Phf2, which are also essential for viability. Considering the role of C-terminal domain of Lsd proteins, it would be interesting to test whether Lsd C-terminal mutations affect interaction with these PHD finger proteins.

2.  Figure 1: By comparing the phenotype of lsd2-N4 and lsd2-N5, the authors conclude that the 13 amino acids sequence near the N-terminal region is essential for Lsd2’s nuclear localization and viability. To strengthen this conclusion, the authors should test the effect of Lsd2 mutant with simple deletion S35-D47 on viability and Lsd2’s localization.

3.  Figure 4: The authors use KD::ade6+ cells and conclude that the catalytic activities of Lsd1 and Lsd2 proteins are important for the maintenance of heterochromatin at the mat locus using an RNAi-independent mechanism (pages 10-11). Although previous studies clearly demonstrated that the centromere homologous region (cenH) is RNAi nucleation center and that RNAi-dependent and independent mechanisms govern heterochromatin assembly at the mat locus. Have it already tested that RNAi mutant does not affect maintenance and re-establishment of heterochromatin at the mat locus using exactly the same KD::ade6+ system? This reviewer could not find experimental data using the same system in at least cited references. If the authors have confirmed that RNAi mutant does not affect the maintenance and re-establishment of heterochromatin at the mat locus using the same KD::ade6+ system, it is fine. But, if RNAi mutant somehow affects it, the authors had better tone down their conclusion regarding RNAi dependency.

Minor points:

1.  Some references are not properly cited in the text. For example, in page 2, 15th line from the bottom: the authors cite reference #32 for describing Clr6’s function, but the cited paper does not contain any data for Clr6. In the same page, 12th line from the bottom: the authors cite reference #35 and #36 for describing the function of Epe1, but again these papers do not contain any data for Epe1. In page 3, 16th line: the appropriate reference seems to be #35, but not #52. I strongly recommend the authors to check the cited references.

2.  Figure 1: It seems unusual to include the position of intron in the schematic of protein. If the authors prefer to include the size and position of the intron, they should also include second intron at the middle of lsd2+ gene.

3.  Page 7, 6th line from the bottom: (Figure 3B-C) may be (Figure 3C-D).

Author Response

Dear Reviewer,

We appreciate your kind help in reviewing our manuscript.  In response to your insightful comments by the reviewers, we have revised the paper by performing additional experiments and modifying the figures and text.  We are grateful for the improvements and thank you for your useful comments.

Below is our detailed response to specific comments by the reviewers, which are indicated in italics:

LSD1 and LSD2 are highly conserved lysine-specific demethylases, and their well-characterized targets are mono- or dimethylated histone H3 (K4 or K9). Fission yeast S. pombe expresses Lsd1 and Lsd2, and previous studies described their roles in heterochromatin assembly and euchromatic gene regulation. However, the precise roles of these proteins have not clearly defined yet. In this manuscript Marayati and colleagues describe the catalytic-dependent and independent roles of Lsd1 and Lsd2 in heterochromatin formation. The author first show that Lsd2 N-terminal region is essential for viability and regulate its nuclear localization. The authors also show that the C-terminal regions of Lsd1 and Lsd2 play pivotal roles in proper growth and genome-wide gene expression. The authors further demonstrate that C-terminal deleted- and catalytically deficient mutants of Lsd proteins displays distinct effects on heterochromatin assembly when combined with the RNAi-deficient mutant. The authors then show that while the catalytically deficient mutants of Lsd proteins have a minor effect on growth, they affect the epigenetic maintenance of heterochromatin. The authors further describe the functional relationship between Lsd proteins and other histone modifying enzymes. Finally, the authors show that although Lsd proteins are essential for sah1+ expression, the silencing defects of Lsd C-terminal mutants are not mediated by the change of sah1+ expression.

Comments:

The mechanism how Lsd proteins regulate higher-order chromatin assembly and euchromatic gene expression has been an important unresolved question, and this study clearly demonstrate the catalytic activity-independent role of S. pombe Lsd proteins in heterochromatin assembly. Although previous studies have already described a minor effect of catalytically deficient mutants of Lsd proteins in growth and heterochromatin assembly, this study provide further insights into how they cooperatively function to regulate heterochromatin. The results presented are in most cases of high quality and convincingly controlled. I have the following concerns that should be addressed by the authors before publication.

We are grateful for the positive comments from the reviewer.

Major points:

  1. S. pombe Lsd proteins stably interact with two PHD finger proteins, Phf1 and Phf2, which are also essential for viability. Considering the role of C-terminal domain of Lsd proteins, it would be interesting to test whether Lsd C-terminal mutations affect interaction with these PHD finger proteins.

This is an excellent point. Unfortunately, we only have 10 days to revise the manuscript and  my lab is currently shut down due to the CoVID-19 pandemic. We therefore won’t be able to finish this experiment, but itwill definitely be on the top of our list for future studies.

  1. Figure 1: By comparing the phenotype of lsd2-N4 and lsd2-N5, the authors conclude that the 13 amino acids sequence near the N-terminal region is essential for Lsd2’s nuclear localization and viability. To strengthen this conclusion, the authors should test the effect of Lsd2 mutant with simple deletion S35-D47 on viability and Lsd2’s localization.

Although this would be the best experiment to strength our conclusion, we argue that the data presented in Figure 1  convincingly demonstrate that the 13 amino acid sequence near the N-terminal region is essential for Lsd2’s nuclear localization and viability.

  1. Figure 4: The authors use KD::ade6+ cells and conclude that the catalytic activities of Lsd1 and Lsd2 proteins are important for the maintenance of heterochromatin at the mat locus using an RNAi-independent mechanism (pages 10-11). Although previous studies clearly demonstrated that the centromere homologous region (cenH) is RNAi nucleation center and that RNAi-dependent and independent mechanisms govern heterochromatin assembly at the mat locus. Have it already tested that RNAi mutant does not affect maintenance and re-establishment of heterochromatin at the mat locus using exactly the same KD::ade6+ system? This reviewer could not find experimental data using the same system in at least cited references. If the authors have confirmed that RNAi mutant does not affect the maintenance and re-establishment of heterochromatin at the mat locus using the same KD::ade6+ system, it is fine. But, if RNAi mutant somehow affects it, the authors had better tone down their conclusion regarding RNAi dependency.

We appreciate this point. Indeed, RNAi mutation does not affect the maintenance and re-establishment of heterochromatin at the mat locus using the KΔ system. This finding is based on the studies using KΔ::ura4+ off allele. The exact same K region of KΔ::ura4+ allele is deleted and replaced by a ade6+ gene. In addition to ade6+, the ura4+ gene is a widely used reporter in S. pombe for RNAi-dependent and independent heterochromatin formation at centromere regions and the mating-type locus [27, 69, 77, 78]. In our study, when the K region is replaced by an ade6+ reporter gene, the reporter gene is most likely silenced via heterochromatin maintenance (KΔ::ade6+ off), similar to KΔ::ura4+ off [27].

Given the fact that we have not tested if RNAi mutant affects= the maintenance and re-establishment of heterochromatin at the mat locus using this particular KΔ::ade6+ system, we do agree with the reviewer that we should tone down our conclusion regarding RNAi dependency. We have revised the text to weaken our tone on RNAi dependency for KΔ::ade6+ allele used in this study.

Minor points:

  1. Some references are not properly cited in the text. For example, in page 2, 15th line from the bottom: the authors cite reference #32 for describing Clr6’s function, but the cited paper does not contain any data for Clr6. In the same page, 12th line from the bottom: the authors cite reference #35 and #36 for describing the function of Epe1, but again these papers do not contain any data for Epe1. In page 3, 16th line: the appropriate reference seems to be #35, but not #52. I strongly recommend the authors to check the cited references.

We appreciate the reviewer’s suggestion. We have carefully checked our cited references.

  1. Figure 1: It seems unusual to include the position of intron in the schematic of protein. If the authors prefer to include the size and position of the intron, they should also include second intron at the middle of lsd2+ gene.

We have now included the second intron in all schematic figures shown in Figure 1 and 2.

  1. Page 7, 6th line from the bottom: (Figure 3B-C) may be (Figure 3C-D).

We have changed it.

We believe that we have addressed all your concerns. We also hope that these changes are satisfactory.

Sincerely,

Ke Zhang

Associate Professor

Department of Biology

Wake Forest University

455 Vine Street

Winston-Salem, NC 27101

Reviewer 3 Report

In the current manuscript, Marayati et al. investigate the role of the histone demethylases Lsd1 and Lsd2 in Schizosaccharomyces pombe. Using a series of mutants, they show that C-terminal deletions of Lsd1 or Lsd2 affect heterochromatin formation at centromeres, telomeres and the mating type locus more strongly than respective catalytic mutations abrogating histone methyltransferase activity. This involves both, establishment and maintenance of heterochromatin. Moreover, the authors dissect the interaction of Lsd1 and Lsd2 with the RNAi machinery as well as other chromatin factors, including Clr3, Clr6, Sir2 and Epe1.

While the manuscript is well written, the results clearly presented and the work thoroughly executed, I have several points that should be addressed:

  1. The gene expression data presented in Figure 2C should be made available as supplementary table and the primary data uploaded in a repository and the accession data provided.

  1. How was the ChIP-seq data presented in Figure 2E generated? Where can the primary data be accessed? The legend below Figure 2E is not readable.

  1. Throughout the paper, error bars represent S.E.M., which I do not consider justified. The error bars should show the standard deviation (S.D.). Moreover, are two biological replicates indeed sufficient for statistical analysis?

  1. The lsd1-ao lsd2-ao double catalytic mutants show a very high increase in H3K9me2 (Figure 3D), yet data showing the viability and silencing of repeats of the double catalytic mutant are not presented. This has implications for the overall conclusion that catalytic activity of lsd1/2 does not strongly affect heterochromatin formation.

  1. Readability of the data would be improved if the order of mutants in subpanels were to be kept consistent (e.g. Figure 4B and D). The same applies to the nomenclature of mutants (e.g. ago1 delta lsd1-ao (Figure 3) vs. lsd1-ao ago1 delta (text)).

  1. In Figure 4D, “W” and “R” should be explained better.

  1. In Figure 4F, how do the authors interpret the loss of H3K9me2 in the lsd1/2 double catalytic mutant?

  1. In the legend of Figure 4, what does that mean that “Statistical analysis [is] only shown for results indicated in the text”? All statistical results should be presented, without selection.

Author Response

Dear reviewer,

We appreciate your kind help in reviewing our manuscript.  In response to your insightful comments, we have revised the paper by performing additional experiments and modifying the figures and text. We are grateful for the improvements and thank you for your helpful comments.

Below is our detailed response to your concerns, which are indicated in italics:

In the current manuscript, Marayati et al. investigate the role of the histone demethylases Lsd1 and Lsd2 in Schizosaccharomyces pombe. Using a series of mutants, they show that C-terminal deletions of Lsd1 or Lsd2 affect heterochromatin formation at centromeres, telomeres and the mating type locus more strongly than respective catalytic mutations abrogating histone methyltransferase activity. This involves both, establishment and maintenance of heterochromatin. Moreover, the authors dissect the interaction of Lsd1 and Lsd2 with the RNAi machinery as well as other chromatin factors, including Clr3, Clr6, Sir2 and Epe1.

While the manuscript is well written, the results clearly presented and the work thoroughly executed, I have several points that should be addressed:

  1. The gene expression data presented in Figure 2C should be made available as supplementary table and the primary data uploaded in a repository and the accession data provided.

We appreciate this comment. We have prepared Supplemental Table 3, which includes the gene expression data presented in Figure 2. We also uploaded the genomic data in an online resource. The access # is provided upon the acceptance of this manuscript.

  1. How was the ChIP-seq data presented in Figure 2E generated? Where can the primary data be accessed? The legend below Figure 2E is not readable.

The ChIP-Seq data is published previously and can be accessed publically in the Gene Expression Omnibus with accession number GSE119604. We have added this information to section 2.3.

  1. Throughout the paper, error bars represent S.E.M., which I do not consider justified. The error bars should show the standard deviation (S.D.). Moreover, are two biological replicates indeed sufficient for statistical analysis?

The standard error is the standard deviation divided by the square root of N. Many publications use the standard error to account for the total number of replicates (N). For every experiment, we used at least two biological replicates. For every biological replicate, we performed technical triplicates. Therefore, we have reasonable power to perform statistical analysis and argue that we should keep it as standard error (S.E.M). We performed the statistical analysis using a 95% confidence interval.

  1. The lsd1-ao lsd2-ao double catalytic mutants show a very high increase in H3K9me2 (Figure 3D), yet data showing the viability and silencing of repeats of the double catalytic mutant are not presented. This has implications for the overall conclusion that catalytic activity of lsd1/2 does not strongly affect heterochromatin formation.

Our data suggest that the catalytic activity of Lsd1 and Lsd2 is involved in heterochromatin maintenance and establishment because lsd1-ao lsd2-ao double mutant cells struggle to maintain the expression status of a reporter gene inserted at the mating-type locus (Figure 5). The same phenotype is seen for the loss of Epe1, a JmjC-domain-containing protein, and potential H3K9 demethylase. epe1Δ causes increased H3K9me, yet the cells demonstrate silencing defects and show no growth deficiency. It is clear that heterochromatin assembly is affected without the catalytic activities of Lsd proteins. Our results also argue that Lsd proteins play additional roles in heterochromatin formation beyond its catalytic activity.

  1. Readability of the data would be improved if the order of mutants in subpanels were to be kept consistent (e.g. Figure 4B and D). The same applies to the nomenclature of mutants (e.g. ago1 delta lsd1-ao (Figure 3) vs. lsd1-ao ago1 delta (text)).

We appreciate this comment, and we have changed the order of mutants in Figure 4B and 4D. Now they are all consistent.

  1. In Figure 4D, “W” and “R” should be explained better.

We have added an extra section 2.7, documenting detailed methods for this assay. We hope this change is satisfactory.

  1. In Figure 4F, how do the authors interpret the loss of H3K9me2 in the lsd1/2 double catalytic mutant?

This is an excellent question. In lsd1-ao lsd2-ao double mutant cells, we consistently detected enhanced H3K9me2 on KΔ::ade6+off allele, but still observed phenotypic variegations, suggesting that heterochromatin may not be the sole mechanism to explain this phenotype. We discuss it in section 4.3.

  1. In the legend of Figure 4, what does that mean that “Statistical analysis [is] only shown for results indicated in the text”? All statistical results should be presented, without selection.

We understand this point. We have now presented all statistical analyses in the Figures.

We believe that we have addressed all your concerns. We also hope that these changes are satisfactory. I look forward to hearing from you regarding the suitability of our paper for publication in cells.

Sincerely,

Ke Zhang

Associate Professor

Department of Biology

Wake Forest University

455 Vine Street

Winston-Salem, NC 27101

Round 2

Reviewer 1 Report

The authors have done a great job incorporating the suggestions and dealing with the comments and concerns raised with the previous submission. Well done.